# Identification of Active Slip Mode and Calculation of Schmid Factors in Magnesium Alloy

**Lichao Li** [1,2], **Chunjoong Kim** [2] **and Young-Min Kim** [1,3,*]

1  Department of Magnesium, Korea Institute of Materials Science, Changwon 51508, Korea
2  Department of Materials Science and Engineering, Chungnam National University, Daejeon 34134, Korea
3  Advanced Materials Engineering, University of Science and Technology, Daejeon 34113, Korea
*  Correspondence: ymkim@kims.re.kr; Tel.: +82-552-803-537

**Abstract:** Four types of slip systems (basal <a>, prismatic <a>, pyramidal <a>, and pyramidal <a + c>) and two types of twinning (extension twinning {$10\bar{1}2$} and contraction twinning {$10\bar{1}1$}) could be identified in magnesium alloys using scanning electron microscopy (SEM) and electron backscatter diffraction (EBSD). In addition, the Schmid factors (SF) of these slip systems were systematically calculated on the basis of the Euler angle which was obtained in EBSD. The identification of slip systems and calculation of SF can help us to understand the contribution made by each type of slip in the plastic deformation of the material, which is important for understanding the deformation mechanism.

**Keywords:** magnesium alloy; slip system; Schmid factor; Euler angle; twinning

## 1. Introduction

In magnesium (Mg) alloys, the (0001) basal plane is the most closely packed plane, which impels the basal <a> slip to be the easiest slip system. The critical resolved shear stress (CRSS) at room temperature for prismatic <a> slip is an order of magnitude higher than that for the basal slip [1]. However, the CRSS of prismatic <a> slip decreases significantly with the increase in temperature and decrease in grain size [1–4]. The pyramidal <a> slip evinces four independent slip modes, although the types of strain that can be accommodated using it are same as those with prismatic and basal slips combined [5]. According to a previous article [6], Mg–Li alloys demonstrate high ductility owing to the high activity of the pyramidal <c + a> slip while accommodating plastic strain. In general, owing to the large Burgers vector of <c + a> dislocations, the pyramidal <c + a> slip could be a potential slip system in magnesium alloys. With an increase in deformation temperature or decrease in grain size, the <c + a> slip system is activated [7]. Moreover, deformation twinning can also be accomplished under low stress. As reported by Barnett, the CRSS ratio for basal <a> slip, {$10\bar{1}2$} extension twinning, prismatic <a> slip, and pyramidal <c + a> slip is 1:0.7:2:15 [8]. The {$10\bar{1}1$} contraction twinning and prismatic <a> slip correspond to the same slip plane, but with different slip directions.

Active slip modes in deformed polycrystals can be identified with transmission electron microscopy (TEM) using the Burgers vector determination technique, but its use is restricted by limited scanning area and large analysis time [1,8–11]. Xu et al. [12] identified active slip modes in Ti alloys through grain orientation mapping using electron backscatter diffraction (EBSD) and strain mapping using high-resolution digital image correlation (HRDIC). However, HRDIC is not as easy to use as scanning electron microscopy (SEM). Many studies have identified slip systems in Mg by comparing the slip trace observed in SEM with the possible slip directions calculated using Euler angles obtained in EBSD [13–18]. The activation of an individual slip system can also be determined by molecular-scale simulation. Jang et al. [19] investigated the mechanism and criterion for

activation of non-basal slip using a molecular statics simulation on dislocation behaviors in multicomponent Mg alloys. Ding et al. [20] also investigated the activation of <c + a> slip and its role to improve the plasticity of Mg using dispersion-inclusive density functional theory in combination with molecular dynamics simulations, and they found that <c + a> dislocations formed more readily on the pyramidal I plane than on the pyramidal II plane in Mg. However, none of these reports explained the method to calculate possible slip systems using the Euler angle. This study aims to provide an efficient and accurate method for identifying slip systems in Mg alloys with hexagonal crystal structures. In addition, the Schmid factor (SF), used to analyze deformation modes in metals, can also be calculated simultaneously for each possible slip system.

## 2. Methodology

The orientation of the grain can be described by Euler angles ($\varphi1$, $\Phi$, $\varphi2$) using the OIM software. Figure 1 shows an example of a crystal with Euler angles (0°, 0°, 0°) and subsequent rotation to the Euler angles (30°, 45°, 20°). The crystal with (0°, 0°, 0°) was rotated by 30°, 45°, and 20° in the counterclockwise direction around the Z-axis, X-axis, and Z-axis, respectively. The angle between the X- and Y-axes was 120°, and the Z-axis was perpendicular to the X- and Y-axes and parallel to the c-axes of the Mg crystal. As the crystal was rotated, its XYZ coordinate system was rotated simultaneously.

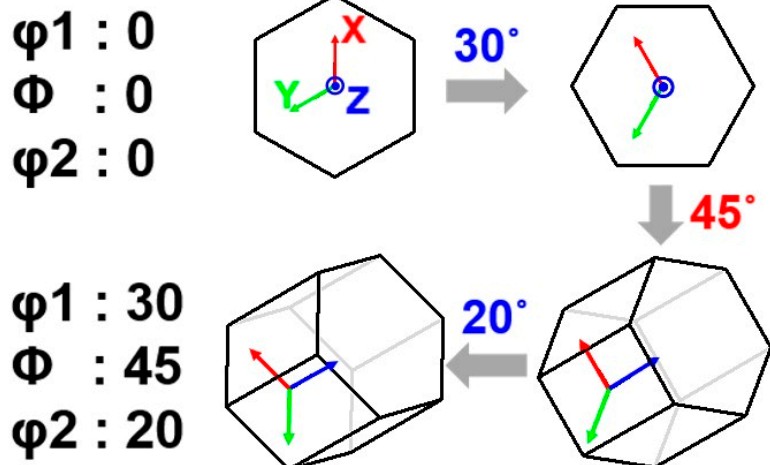

**Figure 1.** A crystal with Euler angles (0°, 0°, 0°) rotated to Euler angles (30°, 45°, 20°).

A four-axis coordinate system is typically used to represent the crystal planes and crystal directions in hexagonal structures, e.g., {*hkil*}–<*uvtw*>. To facilitate the following calculations, a cartesian coordinate system was introduced (as shown in Figure 2). The rotation axes of the X- and Z-axes are denoted as <010> and <001> in the Cartesian coordinate system. For any type of slip plane or twinning plane {*hkil*}, the normal direction of the slip plane in the Cartesian coordinate system can be expressed as follows:

$$[h_1k_1l_1] = [\frac{\sqrt{3}(i-k)}{3}, \frac{2h-k-i}{3}, l(\frac{a}{c})] \tag{1}$$

The slip or twinning direction in the cartesian coordinate system is expressed as

$$[u_1v_1w_1] = [\frac{\sqrt{3}(t-v)}{3}, \frac{2u-v-t}{3}, w(\frac{c}{a})] \tag{2}$$

where *a* and *c* are magnesium lattice constants with values of 0.3209 and 0.5211 nm, respectively (the *c/a* ratio is 1.624) [21].

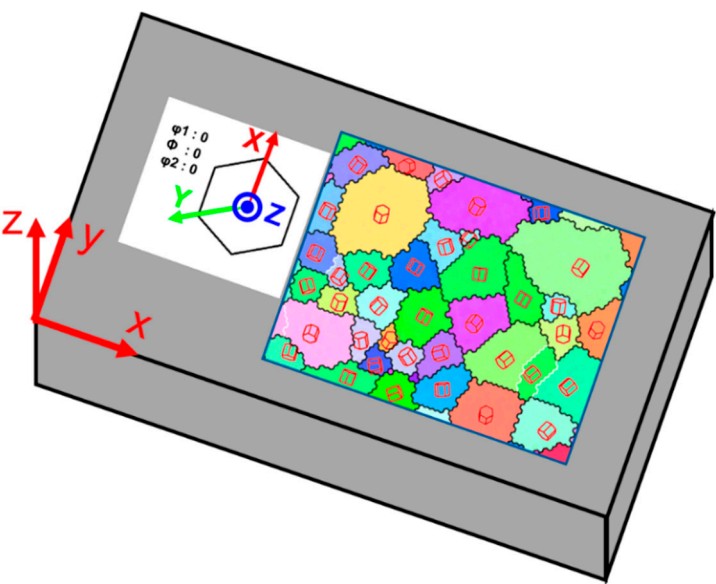

**Figure 2.** Schematic diagram showing the relationship between the crystal with Euler angles (0°, 0°, 0°) and its rotation axis, as well as the cartesian coordinate system. An example of an EBSD inverse pole figure (IPF) map in the cartesian coordinate system is also shown.

Figure 3 shows the schematic relationship among the basal slip plane, normal direction of the basal slip plane, projection of the normal direction, and slip trace on the sample surface. According to the assumption that the dominant slip system is basal slip, the slip trace on the specimen surface remains constant, regardless of the slip direction within the slip plane. The projection of the normal direction of the slip plane has a perpendicular relationship with the slip trace on the sample surface. Therefore, using the grain orientation, the normal direction of any type of slip or twinning plane in the Cartesian coordinate system can be calculated. Furthermore, the direction of the slip trace on the sample surface can be estimated using the calculated normal direction of slip or plane. Additionally, the SF of any slip system can be calculated from the grain orientation if tensile or compressive stresses are applied to the sample. Detailed description of the method to calculate all possible slip traces on the sample surface and the SF of the grain are provided in the next section.

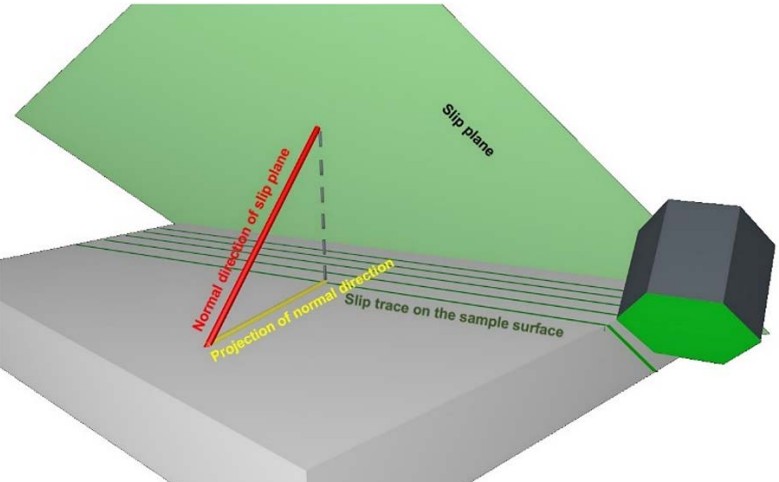

**Figure 3.** Schematic diagram showing the relationship among the slip plane, normal direction of slip plane, projection of the normal direction, and slip trace on the sample. The projection of normal direction on the sample surface has a perpendicular relationship with the slip trace.

### 3. Calculation of Possible Slip Systems and Schmid Factor

*3.1. Calculation of Possible Slip Systems*

The normal direction of the four types of slip systems (basal <a>, prismatic <a>, pyramidal <a>, and pyramidal <c + a>) and two types of twinning (extension twinning {10$\bar{1}$2} and contraction twinning {10$\bar{1}$1}), as well as the slip direction, can be calculated in the Cartesian coordinate system using Equations (1) and (2). It should be noted that, even though contraction twinning {10$\bar{1}$1} and pyramidal <a> have the same slip plane, their morphologies on the sample surface were observed to be different. Prismatic <a> exhibits a number of parallel slip traces on the sample surface, while contraction twinning {10$\bar{1}$1} exhibits a twin-like appearance.

Firstly, the normal direction of the slip plane or slip direction is rotated by $\varphi1°$ around the z-axis. As mentioned before, with the rotation of the crystal, its *XYZ* coordinate system rotates simultaneously. The normal direction of the slip plane or slip direction after rotation around the Z-axis is calculated as follows:

$$x' = x \cdot \cos \varphi_1 - y \cdot \sin \varphi_1. \tag{3}$$

$$y' = x \cdot \sin \varphi_1 + y \cdot \cos \varphi_1. \tag{4}$$

$$z' = z. \tag{5}$$

The *X*-axis <010> and *Z*-axis <001> coordinate systems after rotation around the <001> axis (Z-axis) are $<-\sin \varphi1, \cos \varphi1, 0>$, and <001>, respectively.

Secondly, the normal direction of the slip plane or slip direction is rotated by $\Phi°$ around the $<-\sin \varphi1, \cos \varphi1, 0>$ axis (*x*-axis).

$$x'' = x' \cdot [(-\sin \varphi1)^2 \cdot (1 - \cos \Phi) + \cos \Phi] + y' \cdot [-\sin \varphi1 \cdot \cos \varphi1 \cdot (1 - \cos \Phi)] + z' \cdot (\cos \varphi1 \cdot \sin \Phi). \tag{6}$$

$$y'' = x' \cdot [-\sin \varphi1 \cdot \cos \varphi1 \cdot (1 - \cos \Phi)] + y' \cdot [(\cos \varphi1)^2 \cdot (1 - \cos \Phi) + \cos \Phi] + z' \cdot (\sin \varphi1 \cdot \sin \Phi). \tag{7}$$

$$z'' = x' \cdot (-\cos \varphi1 \cdot \sin \Phi) + y' \cdot (-\sin \varphi1 \cdot \sin \Phi) + z' \cdot \cos \Phi. \tag{8}$$

The *Z*-axis coordinate system <001> after rotation around the $<-\sin \varphi1, \cos \varphi1, 0>$ axis (*X*-axis) is $<\cos \varphi1 \cdot \sin \Phi, \sin \varphi1 \cdot \sin \Phi, \cos \Phi>$.

Thirdly, the normal direction of the slip plane or slip direction is rotated by $\varphi2°$ around the $<\cos \varphi1 \cdot \sin \Phi, \sin \varphi1 \cdot \sin \Phi, \cos \Phi>$ axis (*z*-axis).

$$\begin{aligned}x''' = x'' \cdot &[(\cos \varphi1 \cdot \sin \Phi)^2 \cdot (1 - \cos \varphi2) + \cos \varphi2] \\ &+ y'' \cdot [\cos \varphi1 \cdot \sin \Phi \cdot \sin \varphi1 \cdot \sin \Phi \cdot (1 - \cos \varphi2) - \cos \Phi \\ &\cdot \sin \varphi2] + z'' \\ &\cdot [\cos \varphi1 \cdot \sin \Phi \cdot \cos \Phi \cdot (1 - \cos \varphi2) + \sin \varphi1 \cdot \sin \Phi \cdot \sin \varphi2].\end{aligned} \tag{9}$$

$$\begin{aligned}y''' = x'' \cdot &[\cos \varphi1 \cdot \sin \Phi \cdot \sin \varphi1 \cdot \sin \Phi \cdot (1 - \cos \varphi2) + \cos \Phi \cdot \sin \varphi2] + y' \\ &\cdot [(\sin \varphi1 \cdot \sin \Phi)^2 \cdot (1 - \cos \varphi2) + \cos \varphi2] + z'' \cdot [\sin \varphi1 \\ &\cdot \sin \Phi \cdot \cos \Phi \cdot (1 - \cos \varphi2) - \cos \varphi1 \cdot \sin \Phi \cdot \sin \varphi2.\end{aligned} \tag{10}$$

$$\begin{aligned}z''' = x'' \cdot &[\cos \varphi1 \cdot \sin \Phi \cdot \cos \Phi \cdot (1 - \cos \varphi2) - \sin \varphi1 \cdot \sin \Phi \cdot \sin \varphi2] + y'' \\ &\cdot [\sin \varphi1 \cdot \sin \Phi \cdot \cos \Phi \cdot (1 - \cos \varphi2) + \cos \varphi1 \cdot \sin \Phi \cdot \sin \varphi2] \\ &+ z'' \cdot [(\cos \Phi)^2 \cdot (1 - \cos \varphi2) + \cos \varphi2].\end{aligned} \tag{11}$$

As previously mentioned, the projection of the normal direction of the slip plane on the sample surface has a perpendicular relationship with the slip trace. Then, the slope of the slip trace on the sample surface is calculated as follows:

$$k = -(x_n''' \div y_n'''), \tag{12}$$

where $[x_n''', y_n''', z_n''']$ is the normal direction of the slip plane after the rotation.

### 3.2. Calculation of Schmid Factor

The Schmid factor can be calculated using the following equation [22–24]:

$$m = \cos \Phi \cos \lambda, \tag{13}$$

where $\Phi$ and $\lambda$ are the angles between the loading direction and normal direction of the slip/twinning plane and between the loading direction and slip/twinning direction, respectively. The loading direction $[\sigma 1\ \sigma 2\ \sigma 3]$ can be obtained in the case of application of tensile or compressive stress to the sample. From the previous section, it is clear that the normal direction of the slip or twinning plane and slip or twinning direction of the grain can be calculated. The cosine functions $\cos \Phi$ and $\cos \lambda$ are then calculated as follows:

$$\cos \Phi\,(\lambda) = \frac{\sigma 1 x''' + \sigma 2 x''' + \sigma 3 x'''}{[\sigma 1^2 + \sigma 2^2 + \sigma 3^2]^{1/2} \cdot [x'''^2 + y'''^2 + z'''^2]^{1/2}}, \tag{14}$$

where $[x''', y''', z''']$ is the normal direction of the slip plane or the slip direction after rotation. Unlike dislocation slip, twinning allows simple shear in only one direction, whereas dislocation slip has both forward and backward directions [23,25]. Therefore, the SF ($m$) value for the dislocation slip should be absolute. This variant cannot be formed for a negative value of SF for twinning.

## 4. Case Analysis

A polished Mg specimen was strained by 2% under tensile stress in the loading direction [1 0 0], and then analyzed using SEM and EBSD at the same locations. As shown in Figure 4a, slip traces are clearly visible in one grain. The Euler angles (356.8°, 135.2°, and 250.5°) of this grain could be obtained using EBSD. Figure 4 shows all possible slip systems that could be calculated using the equations mentioned in the previous section. All slip trace slopes ($k$) of the possible slips and corresponding SFs ($m$) are summarized in Table 1. By comparison with the slip trace in the SEM, it can be advocated that the activated slip was a basal slip with an SF value of 0.48.

**Table 1.** All the possible normal directions of slip planes were calculated using Equation (1). The slip trace slops ($k$) and SFs ($m$) with an orientation (356.8°, 135.2°, 250.5°) were calculated in this study.

| Type | Basal &lt;a&gt; | | | | Prismatic &lt;a&gt; | | | |
|---|---|---|---|---|---|---|---|---|
| | Slip plane | Plane normal in cartesian coordinate system &lt;x, y, z&gt; | SF ($m$) | Slop ($k$) | Slip plane | Plane normal in cartesian coordinate system &lt;x, y, z&gt; | SF ($m$) | Slop ($k$) |
| | $(0001)\,[\bar{2}1\bar{1}0]$ | &lt;001&gt; | 0.48 | 17.88 | $(10\bar{1}0)\,[\bar{1}2\bar{1}0]$ | &lt;$-1/\sqrt{3}$, 1, 0&gt; | 0.13 | 3.1 |
| | $(0001)\,[\bar{1}2\bar{1}0]$ | &lt;001&gt; | 0.13 | 17.88 | $(1\bar{1}00)\,[\bar{1}\bar{1}20]$ | &lt;$1/\sqrt{3}$, 1, 0&gt; | 0.25 | −0.69 |
| | $(0001)\,[\bar{1}\bar{1}20]$ | &lt;001&gt; | 0.35 | 17.88 | $(01\bar{1}0)\,[2\bar{1}\bar{1}0]$ | &lt;$-1$, 0, 0&gt; | 0.13 | 0.19 |
| Type | Pyramidal &lt;a&gt; | | | | Pyramidal &lt;a + c&gt; | | | |
| | Slip plane | Plane normal in cartesian coordinate system &lt;x, y, z&gt; | SF ($m$) | Slop ($k$) | Slip plane | Plane normal in cartesian coordinate system &lt;x, y, z&gt; | SF ($m$) | Slop ($k$) |
| | $(10\bar{1}1)\,[\bar{1}\bar{2}\bar{1}0]$ | &lt;$-1/\sqrt{3}$, 1, a/c&gt; | 0.05 | 1.56 | $(11\bar{2}2)\,[\bar{1}\bar{1}23]$ | &lt;$-\sqrt{3}$, 1, 2a/c&gt; | 0.05 | 0.11 |
| | $(1\bar{1}01)\,[\bar{1}1\bar{2}0]$ | &lt;$1/\sqrt{3}$, 1, a/c&gt; | 0.06 | −0.17 | $(\bar{1}2\bar{1}2)\,[\bar{1}2\bar{1}3]$ | &lt;$\sqrt{3}$, 1, 2a/c&gt; | 0.15 | 0.25 |
| | $(0\bar{1}11)\,[2\bar{1}\bar{1}0]$ | &lt;$-2/\sqrt{3}$, 0, a/c&gt; | 0.11 | −0.20 | $(\bar{1}2\bar{1}2)\,[1\bar{2}13]$ | &lt;$-\sqrt{3}$, $-1$, 2a/c&gt; | 0.31 | −0.65 |
| | $(\bar{1}101)\,[\bar{1}\bar{1}20]$ | &lt;$-1/\sqrt{3}$, $-1$, a/c&gt; | 0.39 | −1.23 | $(2\bar{1}\bar{1}2)\,[\bar{2}113]$ | &lt;0, 2, 2a/c&gt; | 0.20 | −0.79 |
| | $(\bar{1}011)\,[1\bar{2}10]$ | &lt;$1/\sqrt{3}$, $-1$, a/c&gt; | 0.17 | 4.39 | $(\bar{2}112)\,[2\bar{1}\bar{1}3]$ | &lt;0, $-2$, 2a/c&gt; | 0.37 | −4.12 |
| | $(01\bar{1}1)\,[2\bar{1}\bar{1}0]$ | &lt;$2/\sqrt{3}$, 0, a/c&gt; | 0.34 | 0.57 | $(\bar{1}\bar{1}22)\,[11\bar{2}3]$ | &lt;$\sqrt{3}$, $-1$, 2a/c&gt; | 0.37 | 1.33 |

**Table 1.** *Cont.*

| Type | Extension twinning {10$\bar{1}$2} | | | | Contraction twinning {10$\bar{1}$1} | | | |
|---|---|---|---|---|---|---|---|---|
| | Slip plane | Plane normal in cartesian coordinate system <x, y, z> | SF (m) | Slop (k) | Slip plane | Plane normal in cartesian coordinate system <x, y, z> | SF (m) | Slop (k) |
| | (10$\bar{1}$2) [$\bar{1}$011] | $< -1/\sqrt{3}, 1, 2a/c>$ | 0.04 | −0.36 | (10$\bar{1}$1) [10$\bar{1}$2] | $< -1/\sqrt{3}, 1, a/c>$ | 0.25 | 1.56 |
| | ($\bar{1}$102) [1$\bar{1}$01] | $< 1/\sqrt{3}, 1, 2a/c>$ | 0.15 | 0.32 | ($\bar{1}$101) [1$\bar{1}$02] | $< 1/\sqrt{3}, 1, a/c>$ | 0.09 | −0.17 |
| | (0$\bar{1}$12) [0$\bar{1}$11] | $< -2/\sqrt{3}, 0, 2a/c>$ | 0.26 | −0.62 | (0$\bar{1}$11) [0$\bar{1}$12] | $< -2/\sqrt{3}, 0, a/c>$ | −0.12 | −0.20 |
| | ($\bar{1}$102) [$\bar{1}$101] | $< -1/\sqrt{3}, -1, 2a/c>$ | 0.24 | −1.81 | ($\bar{1}$101) [$\bar{1}$102] | $< -1/\sqrt{3}, -1, a/c>$ | −0.38 | −1.23 |
| | (10$\bar{1}$2) [10$\bar{1}$1] | $< 1/\sqrt{3}, -1, 2a/c>$ | 0.17 | 5.47 | (10$\bar{1}$1) [10$\bar{1}$2] | $< 1/\sqrt{3}, -1, a/c>$ | −0.41 | 4.39 |
| | (0$\bar{1}$12) [0$\bar{1}$1$\bar{1}$] | $< 2/\sqrt{3}, 0, 2a/c>$ | 0.30 | 0.94 | (0$\bar{1}$11) [0$\bar{1}$12] | $< 2/\sqrt{3}, 0, a/c>$ | −0.30 | 0.57 |

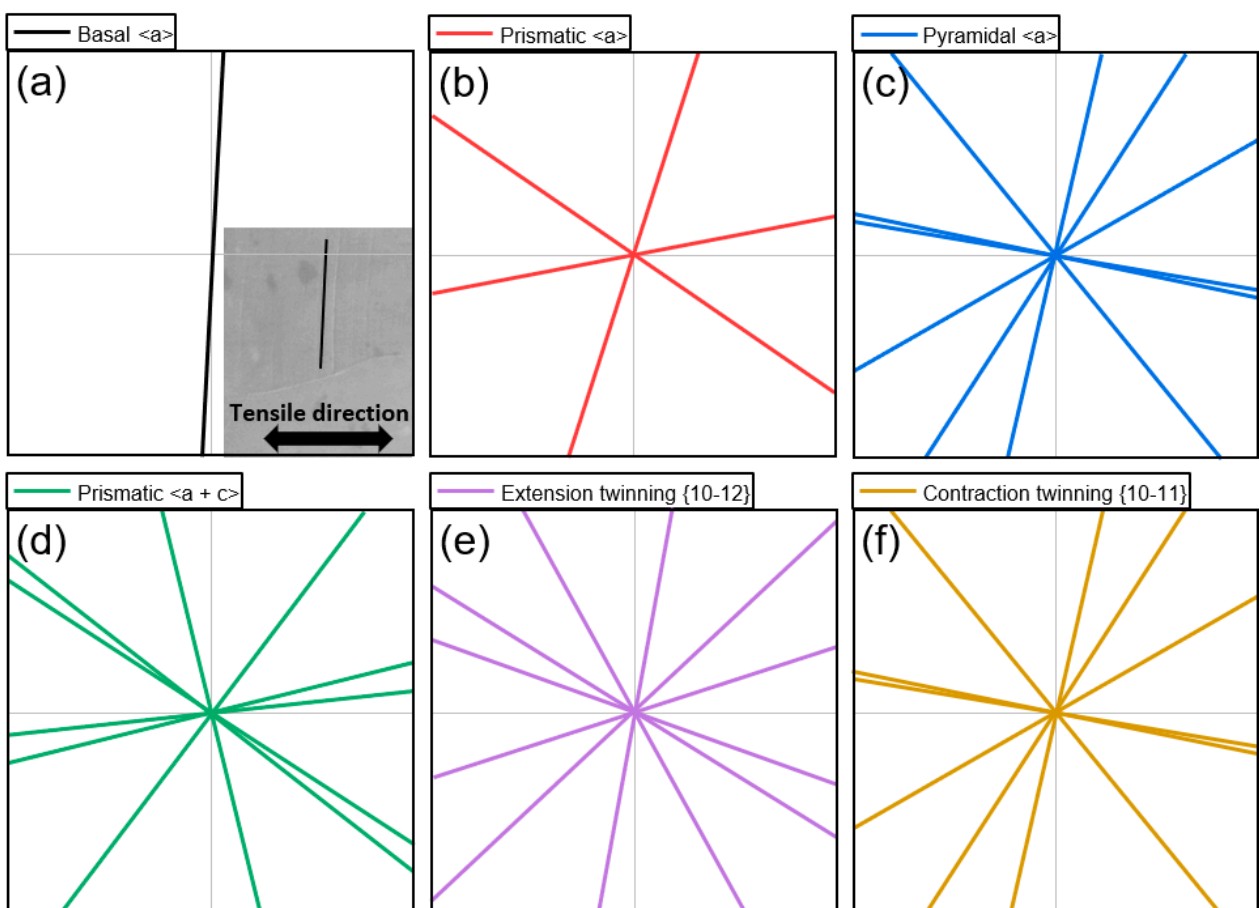

**Figure 4.** All the possible slip traces calculated with the Euler angles (356.8°, 135.2°, 250.5°): (**a**) basal <a> slip; (**b**) prismatic <a> slip; (**c**) pyramidal <a>; (**d**) pyramidal <a + c>; (**e**) extension twinning {10$\bar{1}$2}; (**f**) contraction twinning {10$\bar{1}$1}. Note that (**a**) contains an SEM photomicrograph of basal slip traces.

Figure 5 illustrates an example of this method used to identify extension twinning in deformed Mg alloys. Figure 5a shows the inverse pole figure (IPF) map of the sample, and the Euler angles of grain A were obtained (49.5°, 42.3°, and 265.3°). Figure 5b shows all the possible extension twinning directions. The twinning morphology and possible extension twinning direction match very well. Therefore, this study provides a convenient and accurate method for identifying the twinning styles.

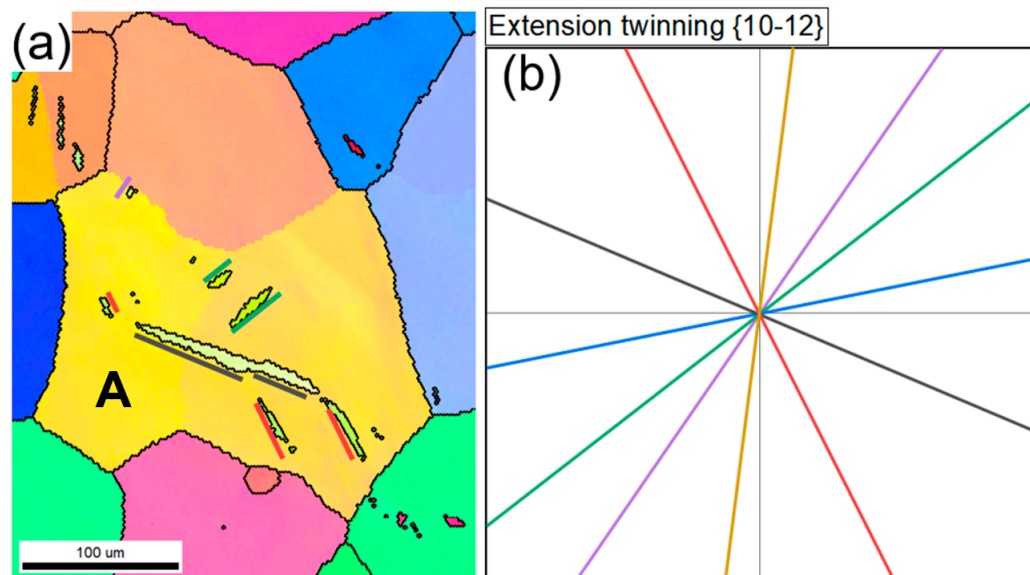

**Figure 5.** An example of identification of extension twinning: (**a**) IPF map of the sample containing extension twinning; (**b**) all possible extension twinning directions calculated with the grain A Euler angles (49.5°, 42.3°, 265.3°). The lines drawn parallel to the extension twins in the grain A in (**a**) correspond to the lines in (**b**) with the same colors and directions.

## 5. Summary

The type of slip system in Mg alloys can be identified using the results of this study. In addition, the corresponding SFs of each possible slip or twinning system can be calculated systematically. A convenient and macroscopic method for analyzing the contribution of individual slip systems during the plastic deformation of magnesium alloys was offered. It is also notable that the method in this study can also be applied to other hexagonal metals; only the *c/a* ratios, such as Cd (1.886), Zn (1.856), Co (1.628), and Ti (1.587), need to be changed for the calculation.

**Author Contributions:** Conceptualization, Y.-M.K.; data curation, L.L.; methodology, L.L.; project administration, Y.-M.K.; supervision, C.K. and Y.-M.K.; writing—original draft, L.L. and Y.-M.K.; writing—review and editing, C.K. and Y.-M.K. All authors have read and agreed to the published version of the manuscript.

**Funding:** This work was supported by This work was supported by the Fundamental Research Program (PNK8390 and PNK8660) of the Korea Institute of Materials Science (KIMS).

**Institutional Review Board Statement:** Not applicable.

**Informed Consent Statement:** Not applicable.

**Data Availability Statement:** Not applicable.

**Conflicts of Interest:** The authors declare no conflict of interest.

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
