# Peer review of "Identification of Active Slip Mode and Calculation of Schmid Factors in Magnesium Alloy"

_metals, doi:10.3390/met12101604_

Round 1
Reviewer 1 Report
The authors presentated a very useful and meaningful tool to identify which kinds of slip systems are responsible for plastic deformation in a polycrystalline material by SEM and EBSD. While there are lots of MD simulations contributing to the properties of <c+a> dislocations in Magnesium, authors might also talk about them in the Intro. part.
Meanwhile, could authors give a statistical result on the proportions of each slip system by a figure? I think that should be a useful information too.
Author Response
1. While there are lots of MD simulations contributing to the properties of <c+a> dislocations in Magnesium, authors might also talk about them in the Intro. part.
Ans.> We agree with the Reviewer. The following sentences and references have been added.
Introduction
(Line 45)… The activation of individual slip system can be also determined by molecular-scale simulation. Jang et al. [19] investigated the mechanism and criterion for activation of non-basal slip using a molecular statics simulation on dislocation behaviors in multicomponent Mg alloys. Ding et al. [20] also investigated the activation of <c+a> slip and its role to improve the plasticity of Mg using dispersion-inclusive density-functional theory in combination with molecular dynamics simulations, and they found out that <c+a> dislocations formed more readily on the pyramidal I plane than on the pyramidal II plane in Mg.
…
References
- Jang, H.-S.;Lee, J.-K.;Tapia, A.J.S.F.; Kim, N.J.;Lee, B.-J. Activation of non-basal <c + a> slip in multicomponent Mg alloys. J. Magnesium Alloys 2022, 10, 585-597, doi:10.1016/j.jma.2021.03.007
- Ding, Z.;Liu, W.;Sung, H.;Li, S.;Zhang, D.;Zhao, Y.;Lavernia, E.J.;Zhu, Y. Origins and dissociation of pyramidal <c+a> dis-locations in magnesium and its alloys. Acta Mater. 2018, 146, 265-272, doi:10.1016/j.actamat.2017.12.049
2. Meanwhile, could authors give a statistical result on the proportions of each slip system by a figure?
Ans.> We would like to clarify that this manuscript mainly introduces a method to identify the type of slip systems. Using the method proposed in this manuscript, the fraction of each slip system in the deformation for rolled Mg alloy sheets will be given in our next paper.
Reviewer 2 Report
The manuscript presents the identification of slip systems and calculation of Schmid factors. This study reports a method to calculate the possible slip systems by Euler angle. It is a topic of interest to the researchers in the related areas. However, the paper needs some improvement before acceptance for publication. My detailed comments are as follows:
1. The references could be improved. The author, for instance, could cite some work about “Calculation of Schmid factors in magnesium”, see for instance: https://doi.org/10.1016/j.scriptamat.2012.05.042.
2. Another obvious problem with this paper is the lack of sufficient experimentation to demonstrate the validity and applicability of the proposed method. Too few experiments make the conclusion of this paper lack persuasive. The author needs to do more experiments with more angles and show them in this paper.
3. It is noted that your manuscript needs careful editing by someone with expertise in technical English editing. paying particular attention to English grammar, spelling, and sentence structure so that the goals and results of the study are clear to the reader.
Overall, it is a nice work and is possible to be published on Metals after fully consideration of aforementioned issues.
Author Response
1. The references could be improved. The author, for instance, could cite some work about “Calculation of Schmid factors in magnesium”, see for instance: https://doi.org/10.1016/j.scriptamat.2012.05.042.
Ans.> We added the above article the Reviewer suggested into References.
References
- Nan, X.-L.; Wang, H.-Y.; Zhang, L.; Li, J.-B.; Jiang, Q.-C. Calculation of Schmid Factors in Magnesium: Analysis of Defor-mation Behaviors. Scr. Mater. 2012, 67, 443–446, doi:10.1016/j.scriptamat.2012.05.042.
2. Another obvious problem with this paper is the lack of sufficient experimentation to demonstrate the validity and applicability of the proposed method. Too few experiments make the conclusion of this paper lack persuasive. The author needs to do more experiments with more angles and show them in this paper.
Ans.> We agree with the Reviewer. The following sentences and Figure 5 have been added at the end of Section 4.
(Line 189) … Fig. 5 illustrates an example of use of this method to identify extension twinning in deformed Mg alloys. Fig. 5(a) shows the inverse pole figure (IPF) map of the sample, and the Euler angle of grain A was obtained (49.5°, 42.3°, and 265.3°). Fig. 5(b) shows all the possible extension twinning directions. The twinning morphology and possible extension twinning direction match very well. Therefore, this study provides a convenient and accurate method for identifying the twinning styles.
…
Figure 5. An example of identification of extension twinning. (a) the IPF map of the sample containing extension twinning; (b) all the possible extension twinning directions calculated with the grain A Euler angle (49.5Ëš, 42.3Ëš, 265.3Ëš).
3. It is noted that your manuscript needs careful editing by someone with expertise in technical English editing. paying particular attention to English grammar, spelling, and sentence structure so that the goals and results of the study are clear to the reader.
Ans.> We have checked and revised some parts where the wrong words were used. In addition, we enlisted the editing services of a professional editing company (Editage) in order to improve the language quality and readability of the manuscript.

Reviewer 3 Report
The paper presents identification of active slip planes and extension and contraction twinning. The calculated slip planes and directions are compared with schematic diagrams (Figures 2, 3 and 4) as could be obtained by SEM and EBSD.
Very important are equations (3)-(13) for calculation of possible slip systems and the Schmid factor.
Author Response
The paper presents identification of active slip planes and extension and contraction twinning. The calculated slip planes and directions are compared with schematic diagrams (Figures 2, 3 and 4) as could be obtained by SEM and EBSD.
Very important are equations (3)-(13) for calculation of possible slip systems and the Schmid factor.
Ans.> We sincerely appreciate the positive comments from the Reviewer.